# Accuracy and efficiency stereo matching network with adaptive feature modulation

**Sen Lin**[☯], **Xinxin Zhuo**[iD][☯]*, **Baozhen Qi**

School of Automation and Electrical Engineering, Shenyang Ligong University, Shenyang, China

☯ These authors contributed equally to this work.
* 2106620725z@gmail.com

**Data Availability Statement:** All relevant data are within the paper.

**Funding:** This paper was supported in part by the National Key Research and Development Program of China under Grant 2018YFB1403303, and in part by the Fundamental Scientific Research Projects

## Abstract

Feature enhancement plays a crucial role in improving the quality and discriminative power of features used in matching tasks. By enhancing the informative and invariant aspects of features, the matching process becomes more robust and reliable, enabling accurate predictions even in challenging scenarios, such as occlusion and reflection in stereo matching. In this paper, we propose an end-to-end dual-dimension feature modulation network called DFMNet to address the issue of mismatches in interference areas. DFMNet utilizes dual-dimension feature modulation (DFM) to capture spatial and channel information separately. This approach enables the adaptive combination of local features with more extensive contextual information, resulting in an enhanced feature representation that is more effective in dealing with challenging scenarios. Additionally, we introduce the concept of cost filter volume (CFV) by utilizing guide weights derived from group-wise correlation. CFV aids in filtering the concatenated volume adaptively, effectively discarding redundant information, and further improving matching accuracy. To enable real-time performance, we designed a fast version named Fast-GFM. Fast-GFM employs the global feature modulation (GFM) block to enhance the feature expression ability, improving the accuracy and stereo matching robustness. The accurate DFMNet and the real-time Fast-GFM achieve state-of-the-art performance across multiple benchmarks, including Scene Flow, KITTI, ETH3D, and Middlebury. These results demonstrate the effectiveness of our proposed methods in enhancing feature representation and significantly improving matching accuracy in various stereo matching scenarios.

## 1 Introduction

Stereo matching [1], a fundamental aspect of computer vision, plays a pivotal role in tasks such as robotics [2], autonomous driving [3], augmented reality [4], and 3D reconstruction [5]. It involves the identification of corresponding points in a pair of stereo images, aiming to establish a reliable depth map that represents the spatial disparities within a scene. The traditional stereo matching methods typically involve four steps: matching cost computation, matching cost aggregation, disparity calculation, and disparity refinement. The primary challenge in stereo matching is indeed to accurately identify corresponding pixels or image patches between

for Higher Education Institutions of the Educational Department of Liaoning Province under Grant LJKMZ20220615. The funders had no role in study design, data collection and analysis, decision to publish, or preparation of the manuscript.

**Competing interests:** The authors have declared that no competing interests exist.

the left and right images. However, real-world scenarios are often complex and changeable. Occlusions occur when objects in the scene obstruct the view of other objects. Changes in lighting conditions can impact the visibility of features, affecting the accuracy of correspondence establishment. These challenges can significantly affect the accuracy of disparity estimation, making stereo-matching a complex task. Accurate disparity estimation is essential for achieving precise depth information, which is crucial in various real-world applications. Inaccuracies in disparity estimation can lead to distorted depth maps, impacting the reliability of subsequent 3D reconstructions.

In recent years, deep learning approaches have made significant advancements in stereo matching by leveraging the power of learning-based feature representations. The integration of convolutional neural networks (CNNs) [6] into stereo matching frameworks has become a common practice to leverage the power of deep learning for improving disparity estimation. However, the complexity of real-world scenarios introduces a set of challenges that demand careful consideration. The integration of CNNs often leads to an increase in computational demands, particularly in scenarios where large and complex datasets are involved. Real-world scenarios are inherently complex, featuring variations in lighting conditions, occlusions, and diverse object geometries. The robust handling of such complexities by CNNs requires careful model design and training strategies to ensure reliable performance across a broad range of scenarios. Failure to adequately address these computational challenges can result in suboptimal stereo matching performance, especially in real-world applications where adaptability to diverse conditions is crucial. GA-Net [7] proposed two novel neural network layers designed to capture local and whole-image cost dependencies. The approach aims to improve stereo matching performance. AANet [8] introduced a sparse points-based intra-scale cost aggregation method to tackle the issue of edge fattening at disparity discontinuities in stereo matching. By utilizing sparse points, the method mitigates the problem and enhances the accuracy of stereo matching. BGNet [9] presented a novel edge-preserving cost volume up-sampling module based on a slicing operation in the learned bilateral grid. This technique improves the quality of disparity estimation by preserving edges in the cost volume. HITNet [10] took a different approach by not explicitly building a volume for disparity estimation. Instead, it employs a fast multi-resolution initialization step, differentiable 2D geometric propagation, and warping mechanisms to infer disparity hypotheses. DecNet [11] addressed the issue of computational costs growing excessively as resolution increases. It proposes densely matching at low resolutions and sparsely matching at higher resolutions to gradually restore lost details in the disparity map. FCStereo [12] introduced a pixel-wise constructive learning approach across viewpoints to enhance the generalization capability of stereo matching. By promoting learning across different viewpoints, the method improves the stereo-matching performance. SRHNet [13] tackled the challenge of dealing with the 4D cubic cost volume used by 3D convolutional filters. It decouples the cost volume into sequential cost maps along the disparity direction, employing a recurrent cost aggregation strategy to handle it effectively. ACVNet [14] presented a novel cost volume construction method that incorporates attention weights generated from correlation clues. These attention weights help suppress redundant information and enhance matching-related information in the concatenation volume, leading to improved stereo matching accuracy. ITSA [15] focused on addressing a factor that hampers network generalization across different domains: shortcut learning. The method explores shortcut connections to improve generalization performance in stereo matching tasks. To better recover fine depth details, CREStereo [16] designed a hierarchical network with recurrent refinement to update disparities in a coarse-to-fine manner and a stacked cascaded architecture for inference.

In the above method, the feature extraction network is mostly based on the feature pyramid module, which expands the receptive field through the convolution of different scales. These methods obtain image features through different convolutional kernels, but this is not clear enough for context features, because capturing context information is not completely equivalent to increasing the size of the receptive field, image context information also includes the reaction and closely related features between pixels. Therefore, capturing contextual information and obtaining many related target information can dynamically reduce the search area. In the stage of feature extraction, DFMNet extends the two-dimensional attention mechanism for stereo matching tasks. Combining spatial and channel attention enhances the matching ability of CNNs. Fast-GFM selects the lightweight deep learning model named MobileNetV2 that can be applied to the feature extraction step in stereo matching, and at the same time adds the attention mechanism SE (Squeeze-and-Excitation) module to solve the important areas ignored in feature extraction. In the construction stage of the cost volume, we build an efficient and accurate cost volume proposal to solve the problem of inefficient utilization of combined volume complementary advantages and effectively improve the matching effect. Fig 1 shows the comparison results of DFMNet and SOTA methods on the KITTI. The matching accuracy of DFMNet is extremely superior. Fig 2 is the result of Fast-GFM on SceneFlow. It has high accuracy in complex details and shows excellent stability. The experimental results provide compelling evidence of the effectiveness of the proposed method. The main contributions of this paper are summarized as follows.

- DFMNet: Introducing the dual-dimensional feature modulation (DFM) block in DFMNet enables capturing broader contextual information in both spatial and channel dimensions. This enhances the feature representation, leading to improved matching performance.

- Fast-GFM: The use of the global feature modulation (GFM) module in Fast-GFM enables a lightweight feature extraction structure. This module enhances the feature expression ability without significantly increasing the model complexity. As a result, the matching accuracy and robustness are improved while maintaining real-time performance.

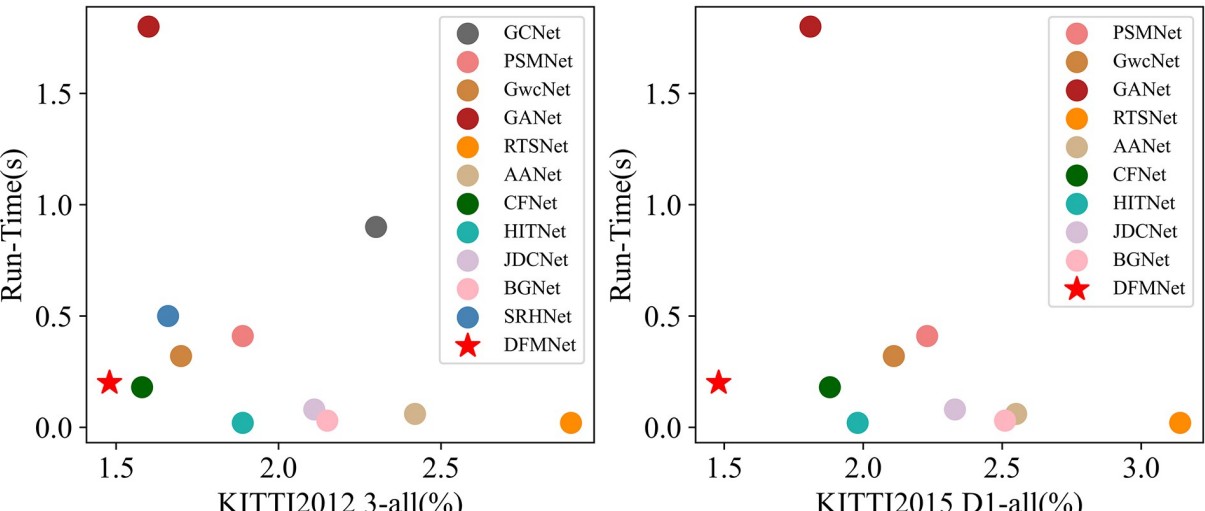

**Fig 1. The comparison of DFMNet with SOTA stereo methods on KITTI 2012 [17]and KITTI 2015 [18].** The term $3 - all$ refers to the percentage of pixels with errors larger than the 3-pixel predictions across all regions, and a lower value is desirable. Similarly, $D1 - all$ represents the percentage of stereo disparity outliers in the first frame across all regions, and a smaller value is preferred.

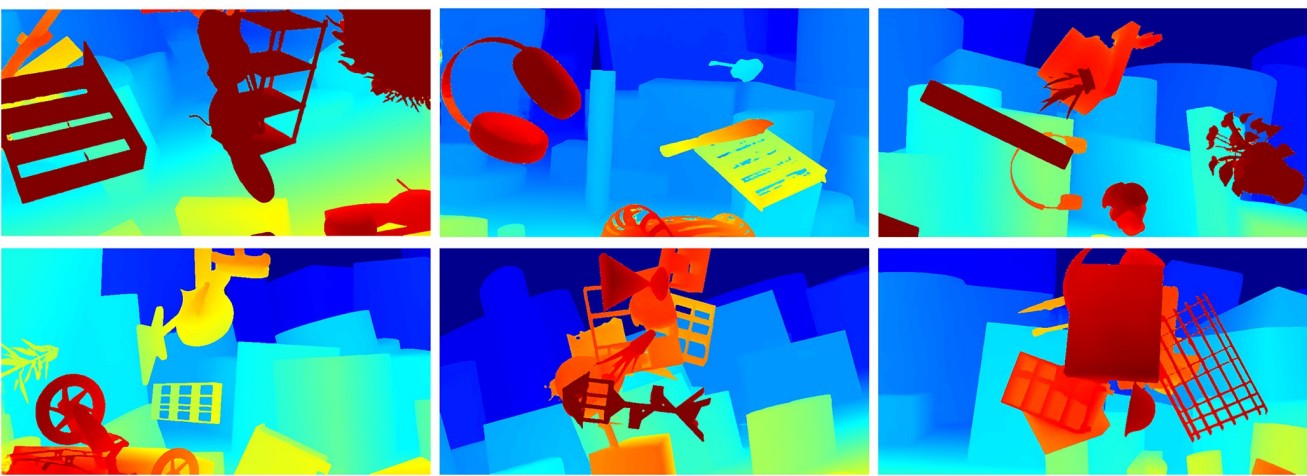

**Fig 2. The prediction of Fast-GFM on Scene Flow [19].**

- Cost Filter Volume (CFV): The incorporation of the cost filter volume effectively suppresses redundant information within the group-wise combined volume. This filtering process enhances the accuracy of the matching process by discarding unnecessary and redundant information.

- Improved Trade-off: DFMNet and Fast-GFM achieve an improved trade-off between matching accuracy and speed. These models outperform state-of-the-art (SOTA) methods on publicly available datasets, demonstrating their effectiveness in enhancing matching performance.

## 2 Related work

### 2.1 Attention mechanism

Attention mechanism has emerged as a crucial concept in deep learning, inspired by the human biological systems that prioritize distinctive parts when processing extensive information. With the advancement of deep neural networks, the attention mechanism has found widespread applications in various domains, effectively reducing computational complexity, and enhancing model performance. SE (Squeeze-and-Excitation) [20], ECA (Efficient Channel Attention) [21], CBAM (Convolutional Block Attention Module) [22], and DA (Dual Attention) [23] are all deep learning models based on attention mechanisms, aiming to enhance the models' perception of important features and improve performance. Subsequently, attention mechanisms have been widely applied in the field of computer vision. In the remote sensing area [24], an effective dual-attention feature extraction network is proposed for feature adaptation. MSPFN [25] for single image rain streak removal employed recurrent calculation to capture the global texture, thus allowing the exploration of the complementary and redundant information at the spatial dimension to characterize target rain streaks.

Overall, incorporating attention mechanisms in feature extraction networks can help improve the performance of various computer vision tasks in robotics and automation. Consequently, DFMNet proposes the integration of a dual-dimension feature modulation block into the convolutional neural network. This block facilitates the extraction of wider contextual

information from different dimensions, enhancing the network's feature representation capabilities and improving its matching performance. Fast-GFM analyzed that the model structure of MobileNetV2 is mainly composed of the Inverted Residual Block and Linear Bottleneck Block. The extraction process often ignores some important areas, so the spatial attention mechanism SE (Squeeze-and-Excitation) module is added to Improve the accuracy and robustness of binocular stereo matching. By leveraging attention mechanisms, the network can selectively focus on crucial features, adapt to contextual cues, and achieve a more effective balance between computational efficiency and accuracy. These advancements contribute to enhanced performance in a broad range of computer vision applications.

## 2.2 Cost volume construction

Cost volume construction is crucial for stereo matching as it directly affects the accuracy and computational efficiency of the disparity estimation process. Researchers continuously explore novel techniques to optimize cost volume construction, including parallel computing strategies, sparse cost volume representations, or learned cost aggregation methods using deep neural networks. DispNetC [19] utilized a correlation layer to perform a vector inner product on the feature maps extracted from the left and right branches, simulating the cost calculation in the standard stereo matching process. GANet [7] and AANet [8] employed the construction of correlation volumes as a crucial component in stereo matching. These methods focus on capturing the relationship between the left and right features at different disparities to estimate accurate disparities. On the other hand, MC-CNN [26], GC-Net [27], and PSMNet [28] took a different approach by connecting the left feature maps with their corresponding right feature maps at each disparity level, forming concat volume. This enables the utilization of both local and global contextual information to improve disparity estimation accuracy. GwcNet [29] introduced the combination of two types of volume to enhance the accuracy of the cost volume and overall stereo matching performance. ACVNet [14] introduced an attention mechanism that selectively highlights informative regions and suppresses redundant information in the cost volume, improving the accuracy of stereo matching.

These various techniques demonstrate the diverse approaches employed in stereo matching to capture and exploit the relationships between stereo images, ultimately advancing the accuracy and robustness of disparity estimation. Matching cost involving the concatenation of features needs to be learned anew through the 3D aggregation network. This typically entails a higher parameter count and computational cost. On the contrary, while the correlation volume offers a means to calculate feature similarity using dot products, it tends to lose significant information. In addressing the limitations of the above approaches, GWC-Net introduces group-wise correlation to construct a more effective combined volume, enhancing the measure of similarity. However, the combined volume merely involves a concat of concatenation and correlation, failing to fully leverage the advantages of both. To address this challenge, we propose a solution in which we extract geometric information from the correlated volume to filter out redundant information in the concatenated volume, yielding a precise and efficient cost filter volume (CFV).

## 2.3 Disparity regression

Disparity regression is indeed the final and essential step in stereo matching process. In recent research, much emphasis has been placed on the construction of cost bodies and cost aggregation, while fewer efforts have been dedicated to enhancing the soft-argmin technique. Initially, GCNet [27] introduced the use of soft-argmin to weight and sum all candidate disparities with their prediction probabilities, leading to a result that satisfies differentiability and achieves

sub-pixel accuracy. However, this method requires the disparity probability distribution to be unimodal. Since then, the construction of 3D cost volume and the soft-argmin disparity calculation have become the foundational framework for mainstream algorithms. Many methods calculate the probability distribution of matching costs at different disparity levels and subsequently use soft-argmin to estimate the expected value of the distribution, which helps predict the final disparity. This approach effectively uses prediction probabilities to weight and sum disparity candidate values when the distribution is unimodal, leading to more accurate results. However, the obtained probability distribution in practical scenarios often exhibits multi-peak phenomena, making the traditional soft-argmin approach less effective. In earlier work, Acf-Net [30] and CDN [31] attempted to improve soft-argmin by training the network with new loss functions to encourage a better unimodal distribution.

In contrast, this paper adopts an enhanced disparity regression strategy, and experimental results demonstrate that this method achieves superior output results. The proposed method represents a step forward in improving the accuracy of disparity estimation, especially in scenarios with multi-peak probability distributions, where the soft-argmin technique may not perform optimally.

## 3 Methods

In the realm of CNN-based stereo matching, considerable strides have been made by researchers in enhancing the accuracy and efficiency of disparity estimation. However, a persistent challenge lies in effectively handling ill-posed areas, particularly those marked by specular reflections or occluded regions. In response to this challenge, we propose an innovative end-to-end learning approach that incorporates attention mechanisms to augment the capabilities of CNNs. Additionally, we present a visual representation of our methodology in Fig 3,

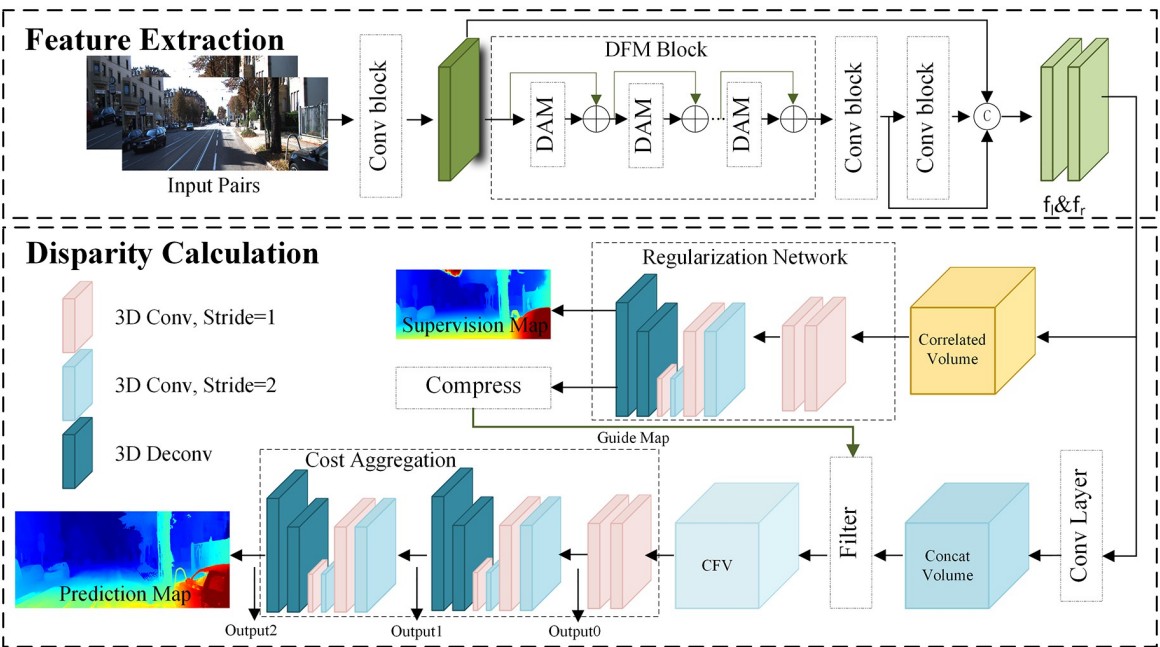

**Fig 3. The architecture of the dual-dimension feature modulation network (DFMNet).** The letter C represents the concatenation operation, while CFV stands for the cost filter volume. DAM, which stands for dual-attention modulation, is a technique that incorporates dual-attention mechanisms into a model to improve performance. DFM Block refers to the dual-dimension feature modulation block. The output of our DFMNet is the prediction map.

delineating four pivotal steps: feature extraction, cost volume construction, cost aggregation, and disparity regression.

## 3.1. DFMNet

**3.1.1 Dual-dimension feature modulation.** By integrating a nuanced comprehension of contextual cues and navigating the intricate interplay between local and global features, we can markedly elevate the precision of matching. This heightened contextual awareness empowers the model to attain a more profound understanding of the overall scene within the input data, thereby resulting in enhanced matching performance.

In Fig 3, the dual-dimension feature modulation (DFM) network is evident, strategically employed to extract rich information from local features obtained through the shallow feature extraction layer. Central to the DFM network is the dual-attention modulation (DAM) block. Given the multifaceted nature of dense prediction tasks, involving variations in scale, orientation, lighting conditions, and occlusions, feature enhancement emerges as a pivotal factor in capturing and accentuating crucial details resilient to such variations. By amplifying the informative and invariant aspects of features, the matching process gains resilience and reliability, facilitating accurate predictions even in challenging scenarios. The efficacy of this approach is underscored by comprehensive empirical results, showcasing a substantial improvement in pixel-level prediction accuracy. In Fig 4, the DAM block is illustrated, comprising two parallel layers: the Spatial Attention Layer (SAL) and the Channel Attention Layer (CAL).

**SAL**. The local feature M is input into the convolution layer to generate maps b, c, and d, respectively. The attention map $S \in R^{N \times N}$ is obtained after matrix operation of b and c, where $N = H/4 \times W/4$ is the number of pixels.

$$S_{ji} = F_{Softmax}(b_i * c_j) \tag{1}$$

where $i, j \in N$.

Meanwhile, the feature maps are updated by aggregating all location features through weighted summation, and the weights determined by the similarity between two locations are to improve each feature regardless of the distance. The final output $E \in R^{C \times H/4 \times W/4}$ of SAL is

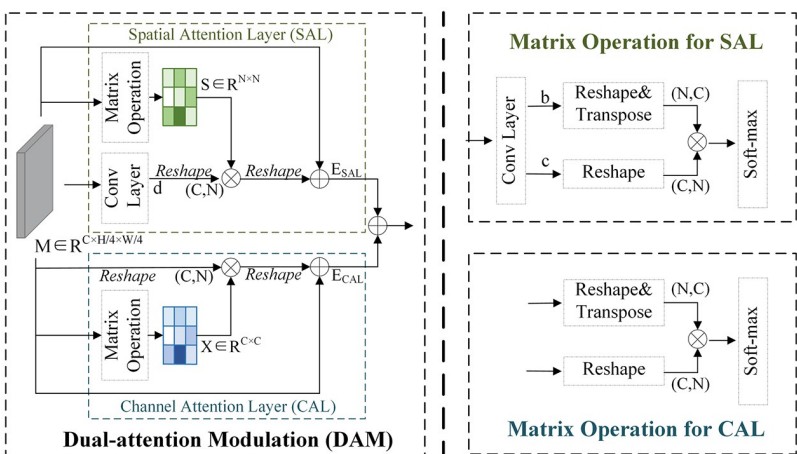

**Fig 4. The structure of dual-attention modulation.**

calculated as follows.

$$E_{SAL(j)} = \alpha \sum_{i=1}^{N} (S_{ji} \cdot d_i) + M_j \tag{2}$$

where $\alpha$ is started from 0.

The Spatial Attention Layer (SAL) plays a pivotal role in capturing spatial context and relationships among local features. It accomplishes this task by deploying attention mechanisms that emphasize relevant spatial regions while suppressing irrelevant or noisy areas. This spatial attention mechanism enables the model to concentrate on crucial regions, filtering out distractions and fostering a deeper understanding of the spatial structure within the input data. Therefore, spatial attention proves instrumental in directing the model's focus towards areas of interest, diminishing reliance on background or irrelevant regions. This attribute becomes particularly valuable in addressing occlusion issues, as the model can prioritize matching the features of the target object without being perturbed by occlusion.

**CAL**. Self-attention is used for the channel attention layer to capture the dependencies between channels. All channels are weighted, and the map of each channel is updated. The final output $E \in R^{C \times H/4 \times W/4}$ of CAL is calculated as follows.

$$X_{ji} = F_{Softmax}(M_i * M_j) \tag{3}$$

$$E_{CAL(j)} = \beta \sum_{i=1}^{C} (X_{ji} \cdot M_i) + M_j \tag{4}$$

where $X_{ji}$ is the channel attention map, $C$ is the input channel number, $i, j \in N$, $\beta$ is started from 0. As shown in Fig 4, before calculating the channel relationship, the convolutional layer is not used to obtain the features of the input data, the reason is to maintain the relationship between different channel maps.

The Channel Attention Layer (CAL) operates at the channel level, aiming to capture interdependencies and dependencies among feature channels. Through the selective amplification or suppression of channel-wise feature responses, CAL enhances the representation of informative channels while attenuating less relevant ones. This channel attention mechanism empowers the model to extract discriminative features effectively and learn robust representations crucial for accurate predictions. CAL enables the model to discern the importance of each channel (feature mapping), contributing to a more nuanced understanding of objects or scene elements at different depths. This refinement enhances the robustness of stereo matching algorithms, ensuring that object boundaries and textures are appropriately considered in the disparity prediction process.

The Spatial Attention Layer (SAL) and Channel Attention Layer (CAL) within the DAM block work collaboratively, operating in parallel to harness both spatial and channel-level contextual cues. Through the integration of these attention mechanisms, the Dual-Dimension Feature Modulation (DFM) block can adeptly leverage global context information. This strategic utilization enhances local features, thereby augmenting the model's proficiency in handling dense prediction tasks.

**3.1.2 Cost volume construction.** For each input stereo pair, the global features $f_l$ and $f_r$ obtained from the from three different levels, namely $l_1$, $l_2$, and $l_3$ ($l_1 = 64$, $l_2 = 128$, and $l_3 = 128$), are compressed to $N \times H/4 \times W/4 (N = 32)$. The concat function is applied to construct concatenated volume.

$$C_{concat}(\cdot, d, x, y) = F_{Concat}(f_l(x, y), f_r(x - d, y)) \tag{5}$$

where $x$ and $y$ is the coordinate of pixel, and $d$ is the disparity level, the size of $C_{Concat}$ is $2N \times D/4 \times H/4 \times W/4$. Geometric information of the stereo pairs extracted from the correlation volume is to obtain guide maps. The unary feature maps with $N_f$ channels ($N_f$=320) are divided into $N_g$ groups ($N_g$=40) from different levels. The $g_{th}$ feature group is denoted as $f_l^g$ and $f_r^g$, and the correlation of diggerent feature group is $C_{ij}^g$. The correlation of different levels $C^l$ is obtained using MAPM [14]. The final correlated volume $C_{corr}$ is constructed as follows.

$$C_{ij}^g(d, x, y) = (f_l^g(x-i, y-j), f_r^g(x-i-d, y-j)) \tag{6}$$

$$C^{l^k}(g, d, x, y) = \frac{1}{N_f / N_g} \sum_{(i,j) \in \Omega^k} w_{ij}^k C_{ij}^g(d, x, y) \tag{7}$$

$$C_{corr} = F_{Concat}(C^{l^1}, C^{l^2}, C^{l^3}) \tag{8}$$

where $C_{corr} \in R^{N_g \times D/4 \times H/4 \times W/4}$, $w_{ij}^k$ represents the adaptive learning weights.

Furtherly, the guide weights $G \in R^{1 \times D/4 \times H/4 \times W/4}$ is obtained by compressing the number of channels. It is applied to decrease redundant information in the concatenate volume. The filter volume at channel $i$ is as follows.

$$CFV_{(i)} = G \odot C_{Concat} \tag{9}$$

where $\odot$ denotes the element-wise product, and the guide weights are applied to all channels of the concatenate volume.

**3.1.3 Cost aggregation and outputs.** Fig 3 provides an overview of the incorporation of the hourglass module within the stereo matching framework. The hourglass module comprises essential elements such as rectified linear units (ReLU), 3D convolution with batch normalization, and stacked 3D hourglass networks. These components play a crucial role in aggregating the cost filter volume (CFV). The subsequent stages involve the cost aggregation network, which generates three disparity maps. Within each output module, two 3D convolutions are employed to generate a 1-channel 4D volume. This volume then undergoes an upsampling process, followed by a softmax operation to convert it into a probability distribution along the disparity dimension. The final disparity map is derived through the soft-argmin operation.

**3.1.4 Loss function.** To guide the training and optimization of the model, a loss function is employed. The loss function serves as a measure of the discrepancy between the prediction and ground truth.

$$L = \lambda_{gw} \cdot F_{L_1}(d_{gw} - d^{gt}) + \sum_{i=0}^{i=2} \lambda_i \cdot F_{L_1}(d_i - d^{gt}) \tag{10}$$

where $d_{gw}$ is the supervision map predicted by regression for the multi-level correlated volume, $d^{gt}$ represents the ground truth map, and $\lambda_{gw}$ represents the coefficient for $d_{gw}$. $d_i$ denotes the output $i$ of DFMNet, and $\lambda_i$ represents the coefficient of the $i_{th}$ disparity prediction.

## 3.2. Fast-GFM

Based on CFV, our primary focus was to design a faster stereo matching method by optimizing the network architecture while maintaining performance. To begin, we introduced a feature extraction structure called the Global Feature Modulation (GFM) module. This module allowed us to capture more intricate and discriminative information from the input stereo images. By extracting robust features, we significantly enhanced the accuracy of matching

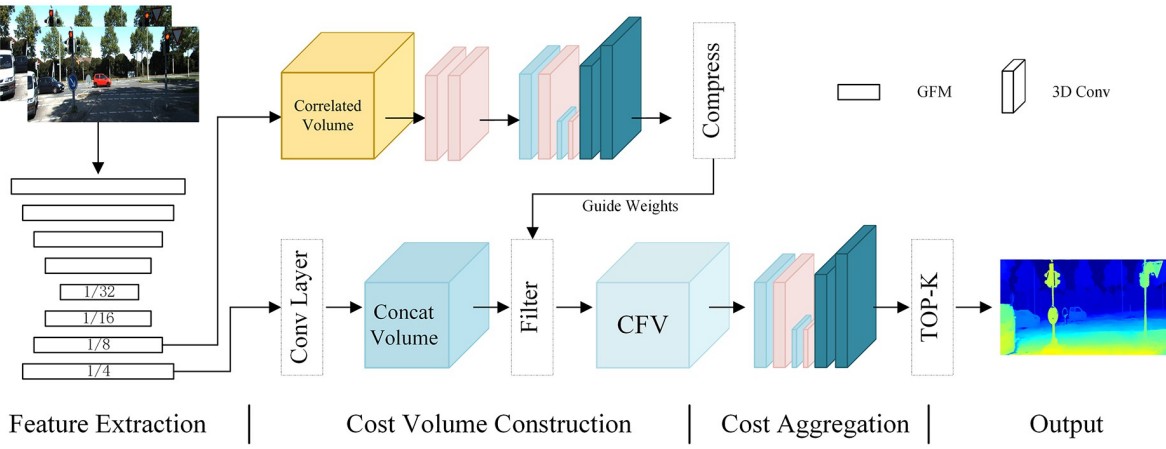

**Fig 5. The structure of Fast-GFM.**

correspondences between the left and right images. This step plays a critical role in establishing reliable depth estimations. We then turned our attention to optimizing the disparity regression process, which is essential for estimating depth information accurately. To achieve this, we implemented the TOP-K disparity regression strategy. This strategy improves both the precision and efficiency of the disparity regression, enabling us to calculate the disparity map with great accuracy. The disparity map represents the pixel-level depth differences between the left and right views. Overall, our proposed method builds upon the strengths of DFMNet. By introducing the GFM module for feature extraction and incorporating the TOP-K disparity regression strategy, we improved the accuracy and efficiency of stereo matching. Moreover, we prioritized light weighting the network architecture to ensure faster inference times without compromising performance. As a result, we have developed a faster stereo matching method as shown in Fig 5.

**3.2.1 Global feature modulation block.** The feature extraction structure is constructed by combining SE block and Convolutional Neural Network (CNN). Consists of 5 parts, and each part consists of several basic blocks of improved MobileNetV2. The output of the first three parts is reduced by 2, 4, and 8 times respectively, and the output of the last two parts is reduced by 16 and 32 times. The model structure of MobileNetV2 is mainly composed of the Inverted Residual Block and the Linear Bottleneck Block, where the inverted residual module is a convolutional block that includes Depthwise Separable Convolution and a 1x1 volume. The multiplication layer significantly reduces the model's parameters and computational load, and the linear bottleneck, an extended version of the inverted residual module, incorporates the Dilated Convolution layer to enhance the receptive field and overall performance. While MobileNetV2 serves as a lightweight deep-learning model for stereo matching feature extraction, certain crucial areas are sometimes overlooked in its feature extraction structure. To address this limitation, we introduce the Squeeze-and-Excitation (SE) module, a spatial attention mechanism. As depicted in Fig 6, the SE Block compresses the input feature map into a feature vector through a global pooling layer and then feeds it into a fully connected layer for learning. The weights learned by this fully connected layer signify the importance of each channel, with channels carrying greater weight contributing more substantially to the network. By multiplying the weight of each channel with its corresponding feature map, a weighted feature map is obtained. This approach directs the network to focus more on important channels,

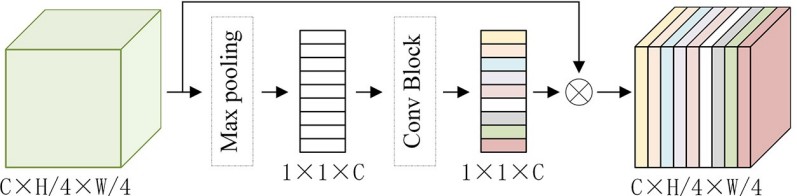

**Fig 6. The structure of SE block.**

thereby enhancing the contribution of information at various positions in the feature map to stereo matching. Without introducing additional complexity to the model, this enhancement boosts the feature expression capability, consequently improving the accuracy and robustness of binocular stereo matching.

**3.2.2 TOP-K.** When dealing with a unimodal disparity distribution, the Soft-argmin can accurately estimate the disparity prediction value. However, in scenarios where the distribution is multimodal or relatively uniform, the expected value of the matching cost may deviate significantly from the actual true value. To address this issue, the Top-K disparity regression method is proposed, which aggregates the cost body of only the top K values per pixel. Specifically, for each pixel, the Top-K regression strategy operates as follows.

1. When *K* is equal to the maximum parallax *d*, the Top-K regression strategy is equivalent to Soft-argmin.

2. When $1 < K < d$, only the first *K* values are used to calculate the estimated disparity value. The first *K* disparity candidate values undergo a soft-max operation, and the final disparity prediction value is obtained as the weighted average of these *K* values, where the sum of the weight is 1. This operation is akin to k-max pooling [32].

3. When *K* = 1, Top-K is equivalent to the argmin operation. In this case, the weight becomes a constant when the parallax is 1, making training impossible. This is the reason Soft-argmin was preferred in previous work.

Moreover, implementing the Top-K regression strategy comes with significant computational costs when calculating the disparity map at full resolution. To mitigate this problem, a design choice is to regress the disparity at 1/4 resolution and then upsample the output predicted disparity map to the original input image resolution. This approach reduces the computational burden while preserving accuracy.

**3.2.3 Loss Function.** The loss function is designed to minimize the disparity error and guide the model to produce accurate disparity predictions during the training process. By using the $L_1$ norm, it penalizes the absolute differences between the predicted disparities and the ground truth disparities. For Fast-GFM, the loss function is defined as follows.

$$L^{Fast} = \lambda_{gw}^{Fast} \cdot F_{L_1}\left(d_{gw}^{Fast} - d^{gt}\right) + \lambda_{gw}^{Fast} \cdot F_{L_1}\left(d^{Fast} - d^{gt}\right) \tag{11}$$

where $d_{gw}^{Fast}$ is obtained through compact guide weights for Fast-GFM, $d^{Fast}$ represents the final output of Fast-GFM, the $\lambda_{gw}^{Fast}$ parameter allows for tuning the importance of each term in the loss function during training.

## 4 Experiments

### 4.1 Datasets

**Scene Flow** [19] is a computer vision dataset used for visual odometry and optical flow estimation tasks. Scene Flow contains many image pairs consisting of Flying Things3D, Driving, and Monkaa, and provides dense ground truth, which is ideal for training and testing stereo matching networks. The collection contains more than 39000 stereo frames in 960$x$540 pixel resolution, rendered from various synthetic sequences. Different from traditional stereo matching datasets, the image pairs in Scene Flow have various degrees of motion and rotation, which makes the stereo matching network need to consider motion information in the matching, to improve the robustness and accuracy of the algorithm. The Scene Flow dataset is commonly used for evaluating the performance of stereo matching methods.

**KITTI** 2012 [17] and KITTI 2015 [18] are two benchmark for computer vision tasks related to autonomous driving. Both datasets were created by the Karlsruhe Institute of Technology and the Toyota Technological Institute in Chicago. KITTI 2012 dataset consists of stereo image pairs, optical flow, and scene flow ground truth, while KITTI 2015 dataset includes stereo image pairs, optical flow, scene flow, and object detection ground truth. KITTI 2015 dataset is an extension of the KITTI 2012 dataset, and it includes additional data and annotations to support more complex computer vision tasks related to autonomous driving. Both datasets have been widely used for research and development of various computer vision algorithms related to stereo matching, optical flow estimation, object detection, and scene understanding in the context of autonomous driving.

**ETH3D** [33] is a benchmark dataset for evaluating the performance of various computer vision algorithms related to 3D reconstruction and image-based modeling. The dataset consists of a collection of high-quality images and their corresponding camera calibration parameters, captured using various cameras and in different environments. ETH3D includes multiple sub-datasets, including structured indoor environments, unstructured outdoor environments, and natural environments. The dataset provides ground truth data for depth maps, camera poses, and point clouds, which can be used for the evaluation and validation of various 3D reconstruction algorithms. Overall, ETH3D is an important benchmark dataset for evaluating and developing computer vision algorithms related to 3D reconstruction and image-based modeling.

**Middlebury** [34] is a benchmark dataset for evaluating the performance of various computer vision algorithms related to stereo matching and optical flow estimation. The dataset consists of a collection of high-quality images and their corresponding ground truth data, captured using calibrated cameras and in different environments. The Middlebury dataset includes multiple sub-datasets, including indoor and outdoor scenes, real and synthetic data, and scenes with varying levels of complexity. The dataset provides ground truth data for disparity maps and optical flow, which can be used for the evaluation and validation of various computer vision algorithms. Overall, the Middlebury dataset is an important benchmark for the evaluation and development of computer vision algorithms related to stereo matching and optical flow estimation.

### 4.2 Implementation details

The overall framework of DFMNet and Fast-GFM are implemented using the PyTorch deep learning library. To optimize the network during the training process, the Adam [35] optimizer utilizes two parameters, $\beta_1$ and $\beta_2$, which control the decay rates of the moving averages of the gradient and its square, respectively. In the context, the values used for these parameters

are $\beta_1 = 0.9$ and $\beta_2 = 0.999$. During both the training and testing, the model is executed on an Nvidia RTX 3090 graphics processing unit (GPU). The specific training and fine-tuning configurations are employed to train and optimize our models.

For DFMNet, the training process involves 35,454 pairs of images from Scene Flow [19]. The batch size is set to 4, and the model is trained for a total of 60 epochs. The initial learning rate is 0.001, and it is decayed by a factor of 2 after the 20th, 28th, 36th, 48th, and 56th epochs. The coefficients of four output are set as $\lambda_{gw} = 0.5$, $\lambda_0 = 0.5$, $\lambda_1 = 0.7$, $\lambda_2 = 1.0$ For Fast-GFM, 35454 pairs from Scene Flow [19] are used as training sets, and the batch size is set to 20. The 28 epochs are trained with 0.001 and decayed by a factor of 2 after epochs 11, 13, 15, 17, and 19. The coefficients of outputs are specified as $\lambda_{gw}^{Fast} = 0.5$, $\lambda^{Fast} = 1.0$. In addition, the pretrained model is fine-tuned for 500 epochs on KITTI 2012 [17] and KITTI 2015 [18]. The initial learning rate is 0.001 and decays by a factor of 10 after the 400th epoch.

## 4.3 The performance of DFMNet and Fast-GFM

**Scene Flow**. As illustrated in Fig 7, our proposed DFMNet surpasses SOTA methods in terms of performance. The experimental results clearly illustrate that DFMNet achieves superior results compared to GANet [7]. DFMNet notably exhibits a remarkable 60.2% reduction in Bad1.0 (Bad Matching Error at 1.0-pixel threshold) compared to GANet [7]. This impressive improvement in performance underscores the effectiveness and superiority of our method. Furthermore, Fig 8 demonstrates that Fast-GFM outperforms the SOTA real-time stereo matching network. This significant improvement provides strong evidence to validate the effectiveness of our method. In summary, our experimental results and comparisons with existing methods show that DFMNet and Fast-GFM achieve remarkable advancements in performance, outperforming state-of-the-art approaches. These findings underscore the efficacy and superiority of our proposed methods in the context of stereo matching.

**KITTI**. As illustrated in Table 1, DFMNet consistently outperforms other methods across various evaluation metrics, demonstrating superior performance in key indicators such as

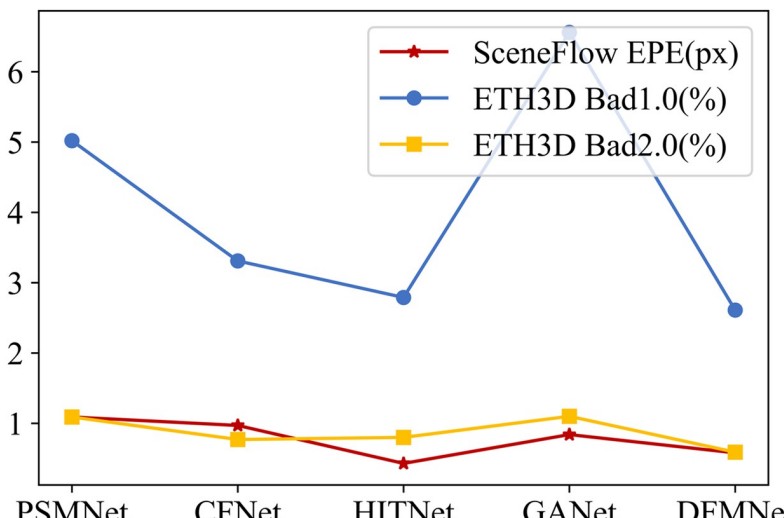

**Fig 7. Evaluation results of DFMNet on Scene Flow [19] and ETH3D [33].** *Bad*1.0 and *Bad*2.0 represent the proportion of pixels whose prediction differs from the ground truth by more than 1.0 and 2.0, respectively. These metrics are used to evaluate the accuracy of the predictions, and lower values indicate better performance.

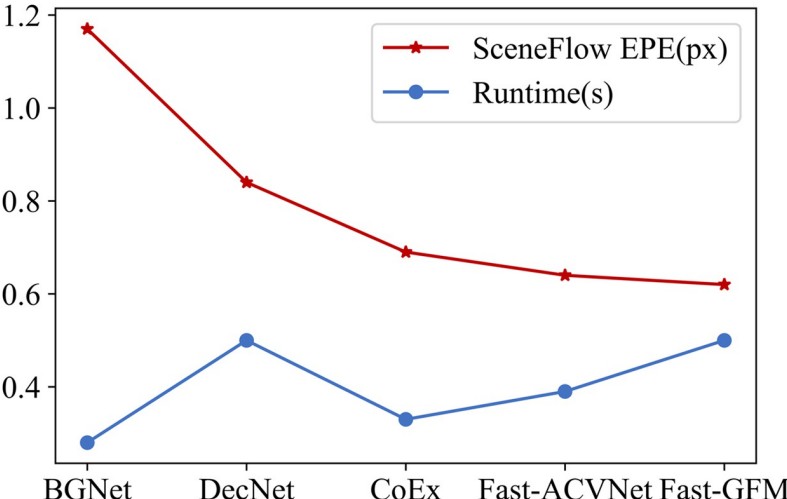

**Fig 8. Evaluation results of Fast-GFM on Scene Flow [19].** *EPE* is used to evaluate the accuracy of the predictions, and lower values indicate better performance.

2-all, 3-all, and EPE-all (px) where it attains the lowest values. Compared to state-of-the-art networks like GCNet, PSMNet, and GwcNet, DFMNet consistently exhibits lower errors and higher accuracy, establishing its excellence in stereo matching tasks. Similarly, Fast-GFM exhibits commendable performance in the assessment of KITTI 2012 and KITTI 2015. It proves to be a strong contender among advanced fast stereo matching networks, showcasing competitiveness against counterparts like RTSNet, AANet, and BGNet. Furthermore, the real-time efficiency of Fast-GFM is noteworthy, with an impressive running time of only 0.05

**Table 1. Quantitative evaluation on KITTI [17, 18].**

| Model | KITTI 2012 | | | | | | KITTI 2015 | | | Time(s) |
|---|---|---|---|---|---|---|---|---|---|---|
| | 2-noc | 2-all | 3-noc | 3-all | EPE-noc(px) | EPE-all(px) | D1-bg | D1-fg | D1-all | |
| GCNet [27] | 2.71 | 3.46 | 1.77 | 2.30 | 0.6 | 0.7 | 2.21 | 6.16 | 2.87 | 0.90 |
| PSMNet [28] | 2.44 | 3.01 | 1.49 | 1.89 | 0.5 | 0.6 | 1.86 | 4.62 | 2.23 | 0.40 |
| GwcNet [29] | 2.16 | 2.71 | 1.32 | 1.70 | 0.5 | **0.5** | 1.74 | 3.93 | 2.11 | 0.32 |
| GANet [7] | 1.89 | 2.50 | 1.19 | 1.60 | 0.5 | **0.5** | 1.48 | - | 1.81 | 1.80 |
| CFNet [36] | 1.90 | 2.43 | 1.23 | 1.58 | **0.4** | **0.5** | 1.54 | 3.56 | 1.88 | **0.18** |
| SRHNet [13] | 2.07 | 3.46 | 1.27 | 1.66 | 0.5 | **0.5** | - | - | - | 0.50 |
| RAFT [37] | 1.92 | 2.42 | 1.30 | 1.66 | **0.4** | **0.5** | 1.58 | 3.05 | 1.82 | 0.38 |
| DFMNet | **1.85** | **2.37** | **1.14** | **1.48** | **0.4** | **0.5** | **1.39** | **3.12** | **1.65** | 0.20 |
| RTSNet [38] | 3.98 | 4.16 | 2.43 | 2.90 | 0.7 | 0.7 | 2.86 | - | 3.14 | 0.02 |
| AANet [8] | 2.90 | 3.60 | 1.91 | 2.42 | 0.6 | 0.6 | 1.99 | 3.96 | 2.55 | 0.06 |
| HITNet [10] | 2.00 | 2.65 | 1.41 | 1.89 | **0.4** | **0.5** | 1.74 | **3.20** | **1.98** | **0.02** |
| BGNet [9] | 3.13 | 3.69 | 1.77 | 2.15 | 0.6 | 0.6 | 2.07 | - | 2.51 | 0.03 |
| BGNet+ [9] | - | - | 1.62 | 2.03 | 0.5 | 0.6 | 1.81 | 4.09 | 2.19 | 0.03 |
| Fast-GFM | **1.94** | **2.41** | **1.39** | **1.86** | 0.5 | **0.5** | **1.72** | 3.50 | 2.07 | 0.05 |

The table in the first half presents the comparison between DFMNet and the precise network method, while the table in the second half displays the comparison between Fast-GFM and the SOTA fast stereo matching network.

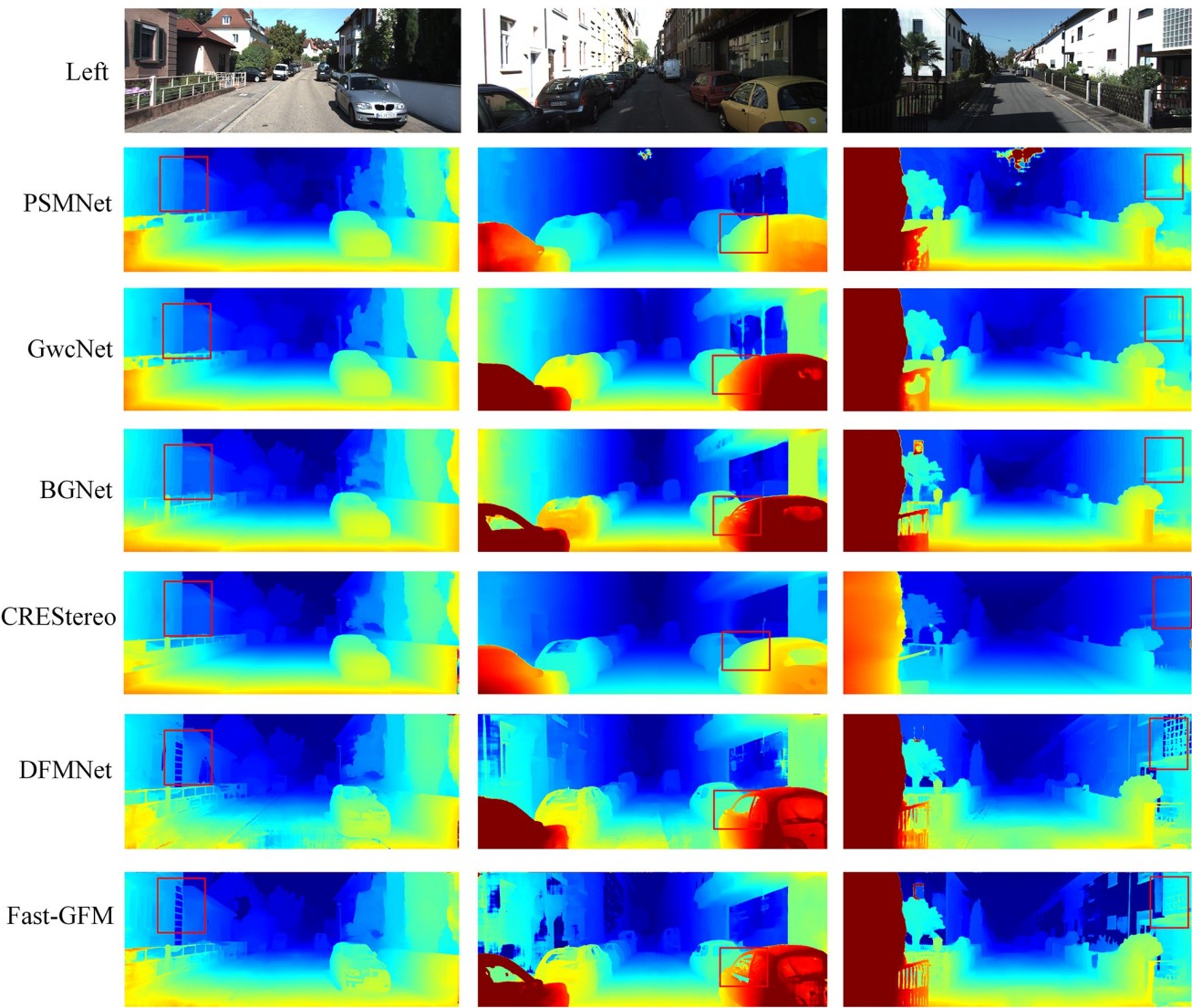

**Fig 9. Results on KITTI 2012 [17].** The matching results of interference area marked in red box.

seconds. The ability of Fast-GFM to deliver high-quality results within such a short timeframe underscores its suitability for real-time applications.

Additionally, as shown in Figs 9 and 10, DFMNet and Fast-GFM exhibit particularly improved performance than SOTA methods like CREStereo [16] in challenging regions such as occluded areas, depth discontinuities, reflections, and other regions marked by the red box. This confirms the effectiveness of DFMNet in handling these difficult scenarios. Overall, the experimental results showcase the effectiveness, efficiency, and superior performance of both DFMNet and Fast-GFM, establishing them as competitive and reliable methods in stereo matching tasks, especially in challenging regions. Their ability to strike a balance between precision and computational speed makes them valuable options for practical real-world applications.

**ETH3D**. As depicted in Fig 7, our proposed DFMNet demonstrates superior performance compared to state-of-the-art stereo matching methods, including HITNet [10] and CFNet

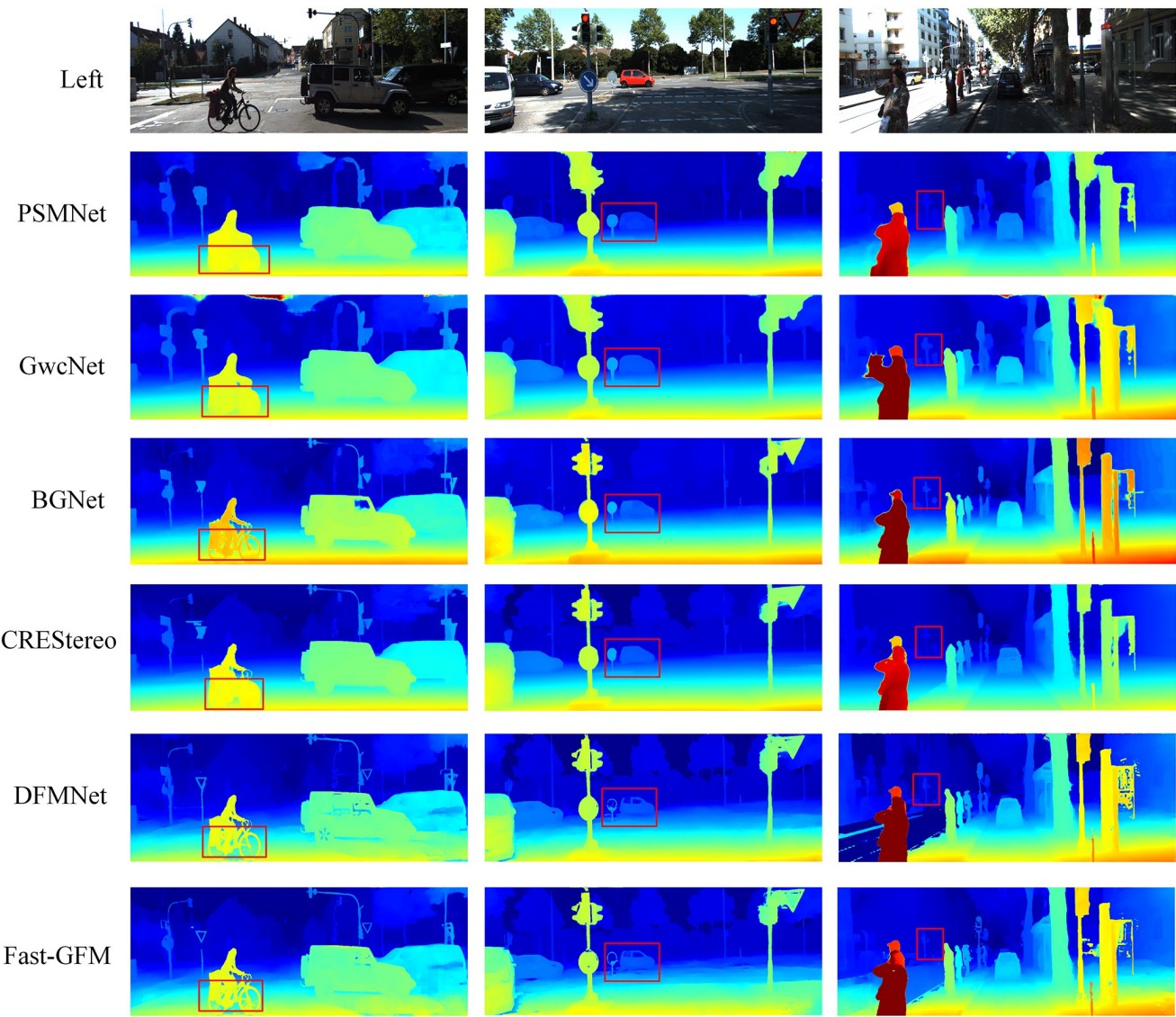

**Fig 10. Results on KITTI 2015 [18].** The matching results of interference area marked in red box.

[36]. The experimental results clearly indicate that DFMNet outperforms these existing methods, showcasing its effectiveness. Furthermore, Fig 11 provides visual evidence of the accurate matching results DFMNet achieved in indoor and outdoor scenes. It can be seen that satisfactory matching results are achieved in complex areas such as occlusion and reflection. These results showcase the superiority of DFMNet in stereo matching tasks and highlight its capability to outperform other state-of-the-art models such as HITNet [10] and CFNet [36]. In Table 2, the performance of Fast-GFM is compared to several other methods. Fast-GFM outperforms the traditional method PatchMatch [39] by an impressive margin of 85.5%. Additionally, it surpasses CFNet by 39.7%, ranking second only to RAFT-Stereo [37] in terms of performance. Overall, our experimental results consistently demonstrate the effectiveness and superiority of DFMNet and Fast-GFM in stereo matching tasks. The impressive performance

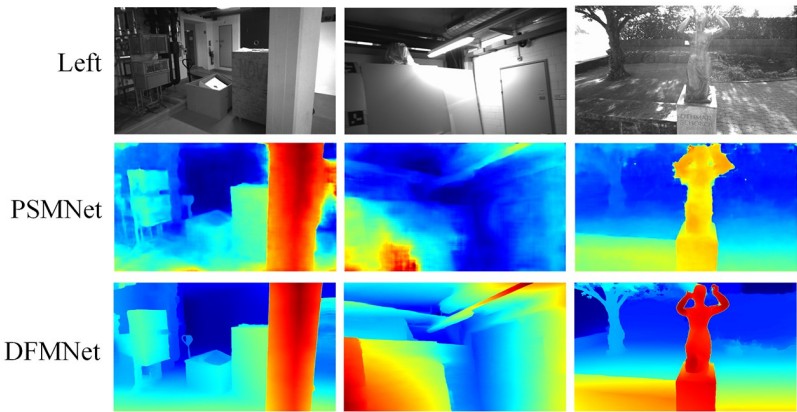

**Fig 11. Results of DFMNet on ETH3D [33].**

gains achieved by our proposed methods compared to state-of-the-art approaches validate their potential for real-world applications in depth estimation and 3D scene reconstruction.

**Middlebury**. As shown in Table 2, on the Middlebury dataset, Fast-GFM exhibited excellent performance, showcasing a significant improvement of 17.2% on half-resolution images and 8% on quarter-resolution images when compared to the RAFT-Stereo [37] method. The impressive performance gains achieved by Fast-GFM highlight its superiority and effectiveness in handling the Middlebury dataset. As shown in Fig 12, details in the foreground and background areas show more precise matching accuracy. These results suggest that Fast-GFM is a competitive and robust method for stereo matching tasks.

## 4.4 Universality of CFV

The proposed cost filter volume (CFV) is a versatile cost volume that can be applied to deep stereo matching networks for most conventional steps. In our experiments, we used PSMNet [28] and GwcNet [29] as baselines and replaced the original volume with our CFV. As depicted in Fig 13, the CFV significantly improves performance. Comparing it with the baselines, the PSMNet-CFV and GwcNet-CFV modules achieve a remarkable 37.3% and 40.3% improvement, respectively, in terms of the average D1 and EPE (End-Point Error). Furthermore, we compared CFV with another widely used cost volume extraction method called Cascaded Cost

**Table 2. Quantitative evaluation of Fast-GFM on ETH3D [33] and Middlebury [34].**

| Model | ETH3D-Bad1.0 | Middlebury-Bad2.0 | |
|---|---|---|---|
| | | half | quarter |
| PatchMatch [39] | 24.1 | 38.6 | 16.1 |
| SGM [40] | 12.9 | 25.2 | 10.7 |
| PSMNet [28] | 10.2 | 15.8 | 9.8 |
| GANet [7] | 6.5 | 13.5 | 8.5 |
| CFNet [36] | 5.8 | 15.3 | 9.8 |
| RAFT-Stereo [37] | 3.2 | 8.7 | 7.3 |
| Fast-GFM | 3.5 | 7.2 | 6.7 |

All methods are trained on Scene Flow. *Bad*1.0 and *Bad*2.0 represent the proportion of pixels whose prediction differs from the ground truth by more than 1.0 and 2.0, respectively.

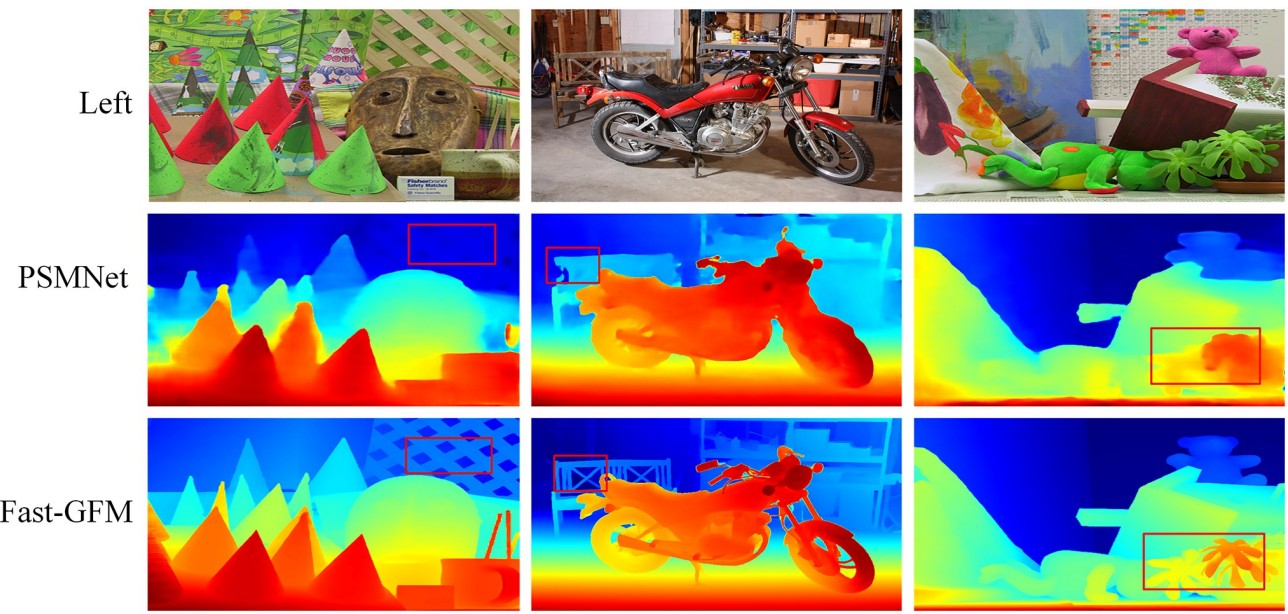

**Fig 12. Results of Fast-GFM on Middlebury [34].**

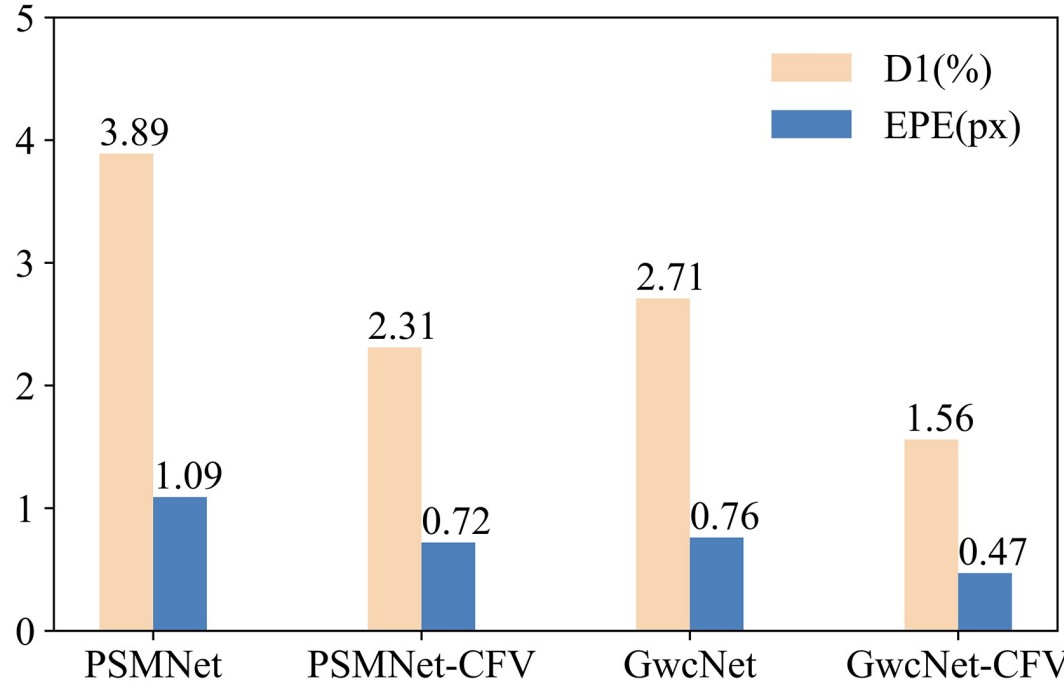

**Fig 13. Universal study of CFV on Scene Flow [19].** *CFV* represents the cost filter volume used in our method. *PSMNet −CFV*, which replaces the concatenated volume with CFV, and *GwcNet−CFV*, which replaces the combined volume with CFV.

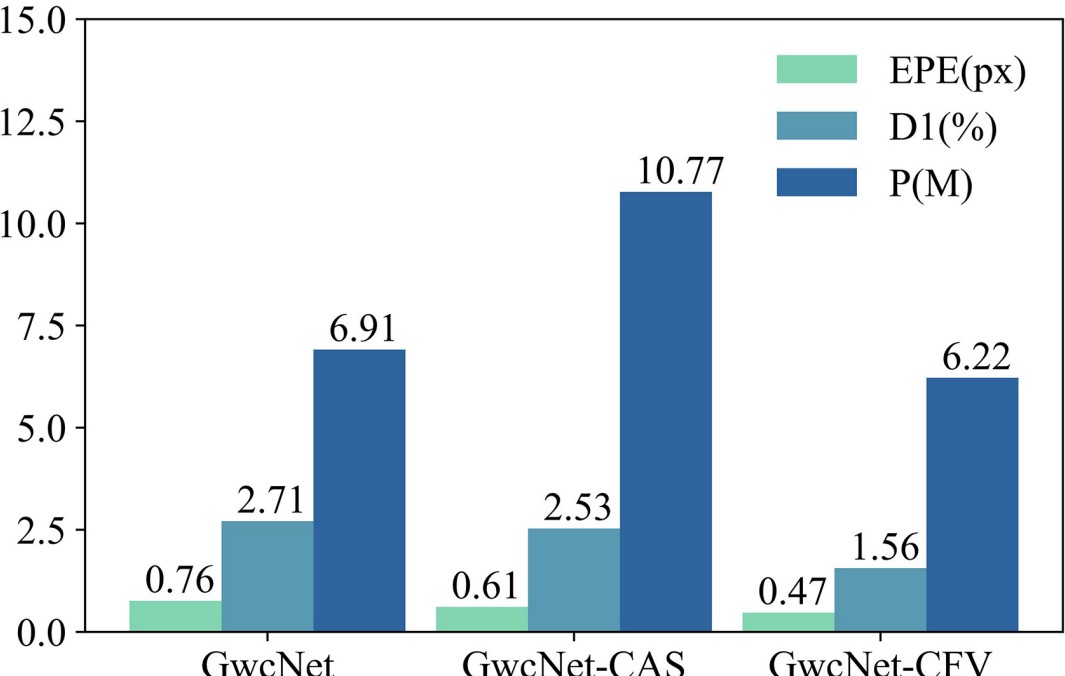

**Fig 14. Comparisons between CFV and CAS [41] on Scene Flow [19].** It can be made by evaluating the performance of two different approaches: *GwcNet−CAS*, which utilizes the Cascaded Cost Volume (CAS) [41] as a replacement for the combined volume, and *GwcNet−CFV*, which replaces the volume with our CFV.

Volume (CAS) [41]. As shown in Fig 14, CFV outperforms CAS. This superiority can be attributed to the fact that CAS directly discards pixel information beyond the predicted disparity range, which may introduce irreversible accumulated errors. On the other hand, CFV adjusts the weights for different disparity levels, allowing it to preserve more information. Importantly, CFV incorporates rich contextual information, which is beneficial to the subsequent aggregation network to some extent.

## 4.5 Effect of Top-K

For disparity regression, we conducted experiments using the Fast-GFM model (i.e. considering the top K most confident predictions) with different Top-K values. As shown in Table 3, D1(%) and EPE are used to evaluate the accuracy of predictions. Furthermore, we observe that the running time of the model gradually increases as the value of Top-K increases. This phenomenon is because fewer candidate values are considered in the process of disparity

**Table 3. Performance of TOP-K when using different K on Scene Flow [19].**

| Model | Top-K | D1(%) | | Runtime(ms) |
|---|---|---|---|---|
| Fast-GFM | 48 | 2.36 | 0.57 | 61 |
| | 32 | 2.43 | 0.61 | 54 |
| | **24** | **2.51** | **0.63** | **50** |
| | 16 | 2.57 | 0.71 | 44 |

*D*1 represents the first frame image mismatch rate.

**Table 4. Analysis of performance with different numbers of hourglasses on Scene Flow [19].**

| Model | CFV | Hourglass Number | Pa (M) | D1 (%) | EPE (px) |
|---|---|---|---|---|---|
| GwcNet [29] | - | 3 | 6.91 | 2.71 | 0.75 |
| GwcNet-CFV | √ | 3 | 7.40 | 1.61 | 0.49 |
| GwcNet-CFV | √ | **2** | **6.22** | **1.68** | **0.51** |
| GwcNet-CFV | √ | 1 | 5.04 | 1.83 | 0.54 |
| GwcNet-CFV | √ | 0 | 3.86 | 2.01 | 0.59 |

The *P* parameter signifies the total number of trainable parameters in the aggregation network, *CFV* represents the cost filter volume, *D*1 represents the first frame image mismatch rate, *EPE* represents network end-point error.

regression when using smaller K values, resulting in less computation. Conversely, larger values of K require more candidate values to be processed, resulting in increased running time. In practical applications, it is crucial to strike a balance between the Top-K value and running time according to task requirements and available computing resources. After careful analysis, we decided to choose a Top-K value of 24 as the final model parameter. The chosen value of Top-K will ensure that the Fast-GFM model can effectively handle multimodal probability distributions and produce reliable disparity estimates.

## 4.6 Hourglass number of aggregation network

We conducted an analysis of the parameters required and the corresponding accuracy in the aggregation network. In this analysis, we used GwcNet [29] as the baseline, which employed three hourglass networks to aggregate the combined volume. We first replaced the original cost volume with our CFV while keeping the other parts constant. Then, we sequentially decreased the number of hourglass networks to evaluate the prediction error based on different numbers of hourglass networks. The results presented in Table 4 demonstrate that the complexity of CFV is slightly higher than that of the combined volume. However, as the number of hourglass networks decreases, the number of parameters in CFV becomes significantly smaller compared to the baseline. Interestingly, even with zero hourglasses, our CFV achieves matching results that outperform GwcNet [29]. This further verifies the high efficiency of our CFV. Considering the trade-off between precision and efficiency, we selected two hourglass structures as our aggregation module. In addition, we conducted an evaluation by using Fast-GFM as a benchmark and sequentially reducing the number of hourglass networks in our aggregation modules. The results presented in Table 5 demonstrate that as the number of hourglass networks decreases, the matching accuracy also declines. However, we also considered the trade-off between accuracy and efficiency during this analysis. After careful consideration, we selected two hourglass structures as our aggregation modules. This decision was

**Table 5. Analysis of accuracy and runtime with different numbers of hourglass structures on Scene Flow [19].**

| Model | CFV | TOP-K | Hourglass Number | D1 (%) | EPE (px) | Runtime (ms) |
|---|---|---|---|---|---|---|
| Fast-GFM | √ | √ | 3 | 1.53 | 0.61 | 56 |
| | √ | √ | **2** | **1.74** | **0.63** | **50** |
| | √ | √ | 1 | 1.96 | 0.67 | 45 |

*CFV* represents the cost filter volume, *D*1 represents the first frame image mismatch rate, *EPE* represents network end-point error. The bold is the final structure of Fast-GFM.

**Table 6. Ablation study of DFMNet on Scene Flow [19].**

| Model | Guide Weights | Hourglass | Supervise Map | >1px (%) | D1 (%) | EPE (px) |
|---|---|---|---|---|---|---|
| GwcNet [29] | - | - | - | 8.03 | 2.71 | 0.76 |
| GwcNet-f | - | - | - | 7.21 | 2.41 | 0.68 |
| GwcNet-f-g | √ | - | - | 6.08 | 1.98 | 0.59 |
| GwcNet-f-g-h | √ | √ | - | 5.63 | 1.77 | 0.51 |
| GwcNet-f-g-h-s | √ | √ | √ | 4.83 | 1.51 | 0.43 |

*f* represents the feature extraction module base on the dual-dimension feature modulation block. *g* represents guide wights. *h* represents the hourglass for guide weights. *s* represents the supervision map. *xpx* represents the percentage of *x* pixel error, *D*1 represents the first frame image mismatch rate, *EPE* represents network end-point error.

based on striking a balance between achieving satisfactory matching accuracy and maintaining efficient computation.

## 4.7 Ablation study

We conducted experiments with various settings, as presented in Table 6. Our baseline model is GwcNet [29]. The results clearly demonstrate that the dual-dimension feature modulation block greatly enhances matching accuracy. It is evident that constructing the filter volume using guide weights yields more accurate results compared to the combined volume. Furthermore, by leveraging the hourglass and supervision map, we achieved even better matching accuracy. The experimental results indicate that GwcNet-f-g-h-s improves by 42.4% and 38.2% over the baseline for D1 and EPE (End-Point Error), respectively. The effectiveness verification of GFM and the TOP-K disparity regression strategy is presented in Table 6, where the GFM improvement index is 5.5%, and the TOP-K disparity regression strategy improvement index is 10.6%. The results in Table 7 provide empirical evidence supporting the effectiveness of these techniques, which can be crucial for making informed decisions about the choice and integration of appropriate methods in the stereo matching pipeline.

## 5 Discussion

In this study, we presented and evaluated two novel methods, DFMNet and Fast-GFM, for stereo matching tasks across various benchmark datasets: Scene Flow, KITTI, ETH3D, and Middlebury. Our findings offer insights into the efficacy, efficiency, and potential real-world applications of these methods. The experimental results showcased remarkable advancements achieved by our proposed methods.

**Table 7. Ablation study of DFMNet on Scene Flow [19].**

| Model | GFM | CFV | TOP-K | >1px (%) | D1 (%) | EPE (px) |
|---|---|---|---|---|---|---|
| GwcNet [29] | - | - | - | 8.03 | 2.71 | 0.76 |
| GwcNet-GFM | √ | - | - | 7.48 | 2.52 | 0.74 |
| GwcNet-CFV | √ | √ | - | 6.47 | 1.96 | 0.69 |
| Fast-GFM | √ | √ | √ | 5.86 | 1.74 | 0.63 |

*GFM* represents the feature extraction module based on the global feature modulation block. *CFV* represents the cost filter volume. *xpx* represents the percentage of *x* pixel error, *D*1 represents the first frame image mismatch rate, *EPE* represents network end-point error.

DFMNet consistently outperformed state-of-the-art approaches such as GANet and CRES-tereo, as evidenced by the significant reduction of 60.2% in Bad1.0 compared to GANet. These improvements were especially pronounced in challenging scenarios marked by occlusions, depth discontinuities, and reflections, highlighting DFMNet's prowess in handling difficult conditions. This aligns with our design goal of feature enhancement, enabling better accuracy in stereo matching. Fast-GFM, on the other hand, exhibited exceptional performance across various datasets. Notably, on the Middlebury dataset, Fast-GFM achieved a substantial improvement of 17.2% on half-resolution images and 8% on quarter-resolution images compared to the state-of-the-art RAFT-Stereo method. These gains underscore Fast-GFM's robustness and its ability to excel even in complex and diverse scenes.

DFMNet's efficiency, without compromising precision, was evident through its runtime being only 0.18s slower than real-time HITNet, and 0.7s faster than GCNet. This balance between accuracy and computational speed is a significant advantage, making DFMNet suitable for real-time applications. In contrast, Fast-GFM achieved an impressive computational speed of 50ms, demonstrating its capability to perform at near-real-time rates. While it sacrifices a marginal amount of accuracy compared to DFMNet, it maintains a higher accuracy than the real-time SOTA stereo matching network.

The effectiveness and efficiency demonstrated by both DFMNet and Fast-GFM have important implications for depth estimation and 3D scene reconstruction in various domains, including robotics, autonomous driving, and augmented reality. These methods have the potential to enhance the accuracy of depth maps, leading to more reliable perception systems in dynamic environments.

## 6 Conclusion

This paper introduces a novel approach called the Dual-Dimension Feature Modulation Network (DFMNet) for stereo matching, aiming to achieve high precision and efficiency. The proposed method incorporates a dual-dimension modulation block to extract information from spatial and channel dimensions. This allows for the combination of local features and global contextual information, resulting in an enhanced representation of the matching features. In addition, the DFMNet utilizes guide weights to filter the concatenated volume. This filtering process effectively removes redundant information, leading to the construction of a high-efficiency cost volume. The matching accuracy is further enhanced by improving the cost volume quality. Furthermore, we designed a fast variant named Fast-GFM based on the global feature modulation (GFM) approach. This variant maintains the effectiveness of the original DFMNet while achieving faster computation times (75% faster), making it a valuable option for real-time applications. The experimental results demonstrate the effectiveness of DFMNet. When compared to GwcNet, DFMNet achieves significant improvements of 19.54%, 21.8%, and 35.5% in the widely used benchmark datasets KITTI 2012, KITTI 2015, and Scene Flow. Our method significantly improves the matching results, showcasing its effectiveness and superiority over existing approaches.

## Acknowledgments

The authors are highly grateful to Editor-in-Chief and the anonymous reviewers for their valuable comments and suggestions to improve the quality of our manuscript.

## Author Contributions

**Conceptualization:** Sen Lin, Xinxin Zhuo.

**Data curation:** Xinxin Zhuo.

**Formal analysis:** Sen Lin, Xinxin Zhuo.

**Investigation:** Sen Lin.

**Methodology:** Sen Lin, Xinxin Zhuo.

**Project administration:** Sen Lin.

**Resources:** Xinxin Zhuo.

**Software:** Sen Lin, Xinxin Zhuo.

**Supervision:** Sen Lin.

**Validation:** Xinxin Zhuo.

**Visualization:** Xinxin Zhuo, Baozhen Qi.

**Writing – original draft:** Sen Lin, Xinxin Zhuo.

**Writing – review & editing:** Sen Lin, Xinxin Zhuo, Baozhen Qi.

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
