## [Decision Letter · Decision Letter 0]

30 Oct 2023

PONE-D-23-27337Accuracy and efficiency stereo matching network with adaptive feature modulationPLOS ONE

Dear Dr. Zhuo,

Thank you for submitting your manuscript to PLOS ONE. After careful consideration, we feel that it has merit but does not fully meet PLOS ONE’s publication criteria as it currently stands. Therefore, we invite you to submit a revised version of the manuscript that addresses the points raised during the review process.

We look forward to receiving your revised manuscript.

Kind regards,

Jean-Christophe Nebel, Ph.D

Academic Editor

PLOS ONE

Journal Requirements:

"This paper was supported in part by the National Key Research and Development Program of China under Grant 2018YFB1403303, and in part by Fundamental Scientific Research Projects of Higher Education Institutions of Liaoning Provincial Department of Education under Grant LJKMZ20220615."

5. Please ensure that you refer to Figure 12 in your text as, if accepted, production will need this reference to link the reader to the figure.

**Additional Editor Comments :**

Please address point by point the comments of the reviewers. There are instances where more details and better justifications of choices are needed.

Reviewers' comments:

Reviewer's Responses to Questions

**Comments to the Author**

1. Is the manuscript technically sound, and do the data support the conclusions?

Reviewer #1: Yes

Reviewer #2: Yes

2. Has the statistical analysis been performed appropriately and rigorously? 

Reviewer #1: Yes

Reviewer #2: Yes

3. Have the authors made all data underlying the findings in their manuscript fully available?

Reviewer #1: Yes

Reviewer #2: Yes

4. Is the manuscript presented in an intelligible fashion and written in standard English?

Reviewer #1: Yes

Reviewer #2: Yes

5. Review Comments to the Author

Reviewer #1: General:

This manuscript is well written and contains comprehensive technical knowledge regarding the stereo matching algorithm to improve the accuracy and efficiency of this algorithm in global and machine learning methods. This author successfully presents the fundamentals of the stereo matching algorithm, the methodology, and experiments to achieve the bad pixel error percentage, and provides extensive analysis and discussion.

Abstract:

In the abstract, the author should explain in detail the interference areas where the mismatching issue occurred.

Introduction:

In the introduction section, the author should explain the fundamentals of stereo matching issues, which are the challenges of stereo correspondence, to give early perspective issues to the readers. The issues consist of the stereo-matching process.

The author also should explain why disparity estimation accuracy is important, especially the impact on the disparity map or 3D reconstruction.

A lack of explanation of the problem statement shows how the integration of CNNs introduces computational challenges and produces problems for complex scenarios.

The author should directly provide the stage in the stereo matching algorithm for the solution or improvement made in this paper.

Figures 1 and 2 are too small; the author needs to resize the figure; the labelling is too small; and KITTI205 should be KITTI2015. These figures also should be explained or summarised in the paragraph, which clearly provides the reader with disparity map accuracy based on quantitative and qualitative

Attention mechanism:

The author can provide a summary of DFMNet and Fast-GFM parameters compared with existing methods in the table.

Cost-volume construction:

Does this stereo matching algorithm use an AGW filter to filter the redundant information? Should explain in detail.

Disparity regression is at what stage?

DFMNet:

The author needs to explain why concatenated and correlated volumes were used in this work and elaborate on the advantages in Figure 3 (and also explain the process flow in Figure 3).

Cost-volume construction:

Explain the type of filter used before the cost aggregation and why this filter is used compared to another established filter.

A lack of explanation on how the CVF is constructed from the correlation and concatenated volume

Disparity refinement:

No explanation of the stage and process to produce the supervision map and the output of DFMNet

Fast-GFM:

The author needs to briefly explain the Top K method and why select regression to the readers.

Figure 5 should label stages and process indicators from matching cost until refinement.

Experiments:

The author should provide and present the computational improvement result when applying the ¼ resolution comparison without reducing resolution at the Top K regression.

Why are the batch sizes set to 4 and 20 and the learning rate = 0.001 and the decay factor of 2? Why is the coefficient lambda set by that? I need to explain that.

Why compared with GANet? I need to explain. How much has improved by DFMNet compared with other SOTAs?

Please also provide the scene flow and ETH3D qualitative results.

Table 1 and 2: is it qualitative or quantitative?

Should we elaborate on Figures 9 and 10 based on stereo correspondence challenges, which are the objectives of this paper?

For the hourglass aggregation network, the author should explain why they used GwcNet as the baseline.

Elaborate on the improvements in Figures 11 and 12 corresponding to the stereo correspondence issues.

Discussion:

There is no evidence or explanation in the results for scenarios for occlusion, depth discontinuities, and illumination variations.

Acknowledgement:

Please write an acknowledgement accordingly.

Reviewer #2: The current work proposes an end-to-end dual-dimension feature modulation network called DFMNet to address the issue of mismatches in interference areas. It utilizes dual-dimension feature modulation (DFM) to capture spatial and channel information separately.

The Fast-GFM needs to be discussed in terms of its prediction time costs compared with other methods.

6. PLOS authors have the option to publish the peer review history of their article (what does this mean?). If published, this will include your full peer review and any attached files.

Reviewer #1: No

Reviewer #2: No

---

## [Author Response · Author response to Decision Letter 0]

1 Mar 2024

Dear Editor:

Thank you for your work on our manuscript (PONE-D-23-27337) titled “Accuracy and efficiency stereo matching network with adaptive feature modulation”. Thanks to the reviewers for their time and their helpful comments. We have carefully read your decision letter and the comments of the two reviewers. All reviewers’ comments have been addressed in our revised manuscript. We resubmit our revised manuscript for further review. Revised parts are marked in red.

We have confirmed our funding information: This paper was supported in part by the National Key Research and Development Program of China under Grant 2018YFB1403303, and in part by the Fundamental Scientific Research Projects for Higher Education Institutions of the Educational Department of Liaoning Province under Grant LJKMZ20220615. The funders had no role in study design, data collection and analysis, decision to publish, or preparation of the manuscript.

We have uploaded the code to: https://github.com/AllaboutJay/DFMNet

The public datasets we used:

(1) Scene Flow:

https://lmb.informatik.uni-freiburg.de/resources/datasets/SceneFlowDatasets.en.html

(2) KITTI: https://www.cvlibs.net/datasets/kitti/

(3) ETH3D: https://www.eth3d.net/

(4) Middlebury: https://vision.middlebury.edu/stereo/data/

We confirm that neither the manuscript nor any parts of its content are currently under consideration or published in another journal. All authors have approved the manuscript and agree with its submission to PLOS ONE. Once again, please accept our thanks for providing us with the opportunity to resubmit our manuscript to PLOS ONE.

Sincerely yours,

Sen Lin, Xinxin Zhuo, Baozhen Qi,

March 1, 2024.

To Reviewer1:

Question1:

In the abstract, the author should explain in detail the interference areas where the mismatching issue occurred.

Response1:

In the revised manuscript, we have included a more comprehensive discussion of the specific interference areas affected by the mismatch, offering a clearer understanding of the challenges and implications of the prediction disparity. Please see the first paragraph of the introduction section. 

Question2:

2.1. In the introduction section, the author should explain the fundamentals of stereo matching issues, which are the challenges of stereo correspondence, to give early perspective issues to the readers. The issues consist of the stereo-matching process.

2.2. The author also should explain why disparity estimation accuracy is important, especially the impact on the disparity map or 3D reconstruction.

2.3. A lack of explanation of the problem statement shows how the integration of CNNs introduces computational challenges and produces problems for complex scenarios.

2.4. The author should directly provide the stage in the stereo matching algorithm for the solution or improvement made in this paper.

2.5. Figures 1 and 2 are too small; the author needs to resize the figure; the labelling is too small; and KITTI205 should be KITTI2015. These figures also should be explained or summarised in the paragraph, which clearly provides the reader with disparity map accuracy based on quantitative and qualitative.

Response2:

2.1 We have revised the introduction to include a more comprehensive explanation of the fundamental challenges inherent in stereo matching. This encompasses a discussion of the key issues, providing readers with a contextual understanding of the complexities involved in the stereo matching process. Please see the first paragraph of the introduction section.

2.2 In the introduction section, we have emphasized the critical significance of accurate disparity estimation in stereo matching, shedding light on its profound impact on subsequent processes such as disparity map generation and 3D reconstruction. Please see the first paragraph of the introduction section.

2.3 We have revised the manuscript to provide a more detailed and explicit elucidation of the challenges associated with the integration of CNNs in stereo matching for complex scenarios. 

Please see the second paragraph of the introduction section.

2.4 We have provided the stage in the stereo matching algorithm for the solution or improvement made in the manuscript. Please see page 3.

2.5. 1)We have resized the figures to ensure improved visibility and legibility.

2) We acknowledge the error in mentioning 'KITTI205', and we have corrected Fig.1 to 'KITTI2015' in the revised manuscript.

3)The content of Figures 1 and 2 has been clearly outlined, please see page 3.

Question3:

The author can provide a summary of DFMNet and Fast-GFM parameters compared with existing methods in the table.

Response3:

We have provided a summary of DFMNet and Fast-GFM parameters compared with existing methods in the table. Please see page 13.

Question4:

Does this stereo matching algorithm use an AGW filter to filter the redundant information? Should explain in detail.

Disparity regression is at what stage?

Response4:

1）Please see Eq 9 for the filter operation, where ⊙ represents the element-wise product, and the guide map is applied to all channels of the initial concat volume.

█(CFV_((i) )=G⊙C_Concat#(9) )

2) We have added a description of Disparity regression, please see section 3.1.3.

Question5:

The author needs to explain why concatenated and correlated volumes were used in this work and elaborate on the advantages in Figure 3 (and also explain the process flow in Figure 3).

Response5:

1)The use of concatenated and correlated volumes in our work is strategically designed to enhance the efficiency and accuracy of the cost volume generation. Please see the second paragraph of section 2.2 for the details.

2) We have added the explanation for Fig.3. Please see page 6.

Question6:

Explain the type of filter used before the cost aggregation and why this filter is used compared to another established filter.

A lack of explanation on how the CVF is constructed from the correlation and concatenated volume

Response6:

1）Please see Eq 9 for the filter operation, where ⊙ represents the element-wise product, and the guide map is applied to all channels of the initial concat volume.

█(CFV_((i) )=G⊙C_Concat#(9) )

2）We propose a solution in which we extract geometric information from the correlated volume to filter out redundant information in the concatenated volume, yielding a precise and efficient cost filter volume (CFV).

3）We have revised the content of the cost volume construction section to increase the readability of the article. Please see pages 8 and 9.

Question7:

No explanation of the stage and process to produce the supervision map and the output of DFMNet

Response7:

1) The supervision map predicted by regression for the multi-level correlated volume. Please see Eq 10 on page 9, where obtained by guide weights.

2) Outputs of DFM are added in section 3.1.3.

Question8:

The author needs to briefly explain the Top K method and why select regression to the readers.

Figure 5 should label stages and process indicators from matching cost until refinement.

Response8:

1) Please see section 3.2.2 for the explanation and why select. 

2) All stage has been labeled.

Question9:

9.1. The author should provide and present the computational improvement result when applying the ¼ resolution comparison without reducing resolution at the Top K regression.

9.2. Why are the batch sizes set to 4 and 20 and the learning rate = 0.001 and the decay factor of 2? Why is the coefficient lambda set by that? I need to explain that.

9.3. Why compared with GANet? I need to explain. How much has improved by DFMNet compared with other SOTAs?

9.4. Please also provide the scene flow and ETH3D qualitative results.

9.5. Table 1 and 2: is it qualitative or quantitative?

9.6. Should we elaborate on Figures 9 and 10 based on stereo correspondence challenges, which are the objectives of this paper?

9.7. For the hourglass aggregation network, the author should explain why they used GwcNet as the baseline.

9.8. Elaborate on the improvements in Figures 11 and 12 corresponding to the stereo correspondence issues.

Response9:

9.1. As mentioned in the manuscript, this is an acceleration strategy, and the accuracy has little impact on the overall network. Fast-GFM focuses on accelerating the network and improving the speed of each component.

9.2. 1) Batch size is a hyperparameter that can be tuned by trying different values. Use cross-validation or validation sets to evaluate model performance at different batch sizes and choose the batch size that performs best.

2)The hyperparameter settings like learning rate in the deep learning network have been optimized and verified multiple times based on previous experience.

9.3. 1)GANet mainly solves the huge calculation problem caused by CNN. The method proposed in this letter builds an accurate and efficient cost volume, which can lightweight the cost aggregation network. Therefore, it can indirectly solve the calculation problem. 

2)The accuracy of the method proposed in this article reaches the SOTA level. Compared with PSMNet, the mismatch rate increased by 25%, compared with CFNet, it increased by 4.69%, and compared with RAFT, all indicators showed a certain degree of improvement.

9.4 Please see the scene flow qualitative results in Fig.2 and ETH3D results in Fig.11.

9.5. The titles of all tables have been checked and corrected.

9.6. Please see the discussion of Fig.9 and Fig.10 in the last paragraph on page 13.

9.7 CFV is an improvement on the combined volume proposed in GWCNet. Taking GWCNet as the benchmark and replacing it with the accurate and efficient CFV proposed in this article can directly demonstrate the improvement effect. We know that GWCNet selected three hourglass structures in the cost aggregation structure. Since we have constructed an accurate and efficient cost volume, theoretically we infer that reducing the number of hourglass structures can also achieve higher matching accuracy. We set up an experiment, reduced the number of hourglass structures in turn, recorded the matching results, and finally found that even if the hourglass structure is not used, the matching results of CFV are still higher than the combined volume.

9.8 We have added the description and summary of Fig.11 and Fig.12. Please see pages 14 and 16.

Question10:

There is no evidence or explanation in the results for scenarios for occlusion, depth discontinuities, and illumination variations.

Response10:

1) The official KITTI data set does not directly focus on the matching performance of specific areas such as occlusion areas or reflective areas. However, we consider that the pixel error in the non-occluded area can indirectly reflect the matching accuracy of the occluded area. The presence of occluded areas may lead to errors in the matching of non-occluded areas, as the algorithm may be affected by occluded areas to produce inaccurate depth estimates. When evaluating pixel error in non-occluded areas, the performance of the algorithm in occluded areas is considered, since inaccurate matching in these areas may affect the overall pixel error.

2). We conducted subjective comparison experiments focusing on specific scenarios to address this concern. The experimental results indicate that our proposed method consistently exhibits more accurate matching results in areas of occlusion and reflection when compared to the state-of-the-art (SOTA) methods shown in Fig.9 and Fig.10.

Question11:

Please write an acknowledgement accordingly.

Response11:

The authors are highly grateful to Editor-in-Chief and the anonymous reviewers for their valuable comments and suggestions to improve the quality of our manuscript.

To reviewer2:

Question1:

The Fast-GFM needs to be discussed in terms of its prediction time costs compared with other methods.

Response1:

We have added the discussion of the prediction time. Please see the first paragraph on page 13.

---

## [Decision Letter · Decision Letter 1]

11 Mar 2024

Accuracy and efficiency stereo matching network with adaptive feature modulation

PONE-D-23-27337R1

Dear Dr. Zhuo,

We’re pleased to inform you that your manuscript has been judged scientifically suitable for publication and will be formally accepted for publication once it meets all outstanding technical requirements.

Kind regards,

Jean-Christophe Nebel, Ph.D

Academic Editor

PLOS ONE